# ABA Mediates Plant Development and Abiotic Stress via Alternative Splicing

**DOI:** 10.3390/ijms23073796

**Published:** 2022-03-30

**Authors:** Xue Yang, Zichang Jia, Qiong Pu, Yuan Tian, Fuyuan Zhu, Yinggao Liu

**Affiliations:** 1State Key Laboratory of Crop Biology, College of Life Science, Shandong Agricultural University, Tai’an 271018, China; xueyang202001@163.com (X.Y.); jiazc973@163.com (Z.J.); ty15610312143@163.com (Y.T.); 2Co-Innovation Center for Sustainable Forestry in Southern China & Key Laboratory of National Forestry and Grassland Administration on Subtropical Forest Biodiversity Conservation, College of Biology and the Environment, Nanjing Forestry University, Nanjing 210037, China; qiongpu2022@163.com; 3College of Mechanical and Electronic Engineering, Shandong Agriculture and Engineering University, Jinan 250000, China

**Keywords:** abscisic acid, alternative splicing, abiotic stress responses, plant development

## Abstract

Alternative splicing (AS) exists in eukaryotes to increase the complexity and adaptability of systems under biophysiological conditions by increasing transcriptional and protein diversity. As a classic hormone, abscisic acid (ABA) can effectively control plant growth, improve stress resistance, and promote dormancy. At the transcriptional level, ABA helps plants respond to the outside world by regulating transcription factors through signal transduction pathways to regulate gene expression. However, at the post-transcriptional level, the mechanism by which ABA can regulate plant biological processes by mediating alternative splicing is not well understood. Therefore, this paper briefly introduces the mechanism of ABA-induced alternative splicing and the role of ABA mediating AS in plant response to the environment and its own growth.

## 1. Introduction

Plant growth and development are significantly affected by environmental factors. Among them, the most important environmental factors are abiotic stresses such as drought and low temperature. Harsh environments affect plant physiology and distribution, especially crops [1]. Abiotic stress triggers a series of systemic responses in plants, and each stress response pathway is not independent. The responses generated by local stress are expressed in local and distal tissues, resulting in the systemic acquired domestication of plants [2]. When plants are subjected to environmental stimuli, such as drought, salt, and extreme temperature, ABA accumulates in different degrees. Therefore, ABA is considered the main plant hormone for plants to respond to external environmental stresses. The ABA signaling pathway is a central reaction pathway for environmental adversities, such as salt and drought in plants [3,4]. 

ABA is one of the most important plant hormones widely distributed in higher plants. It is named for its ability to promote leaf abscission. Under adversity stress, plants accumulate a large amount of ABA, thereby enhancing their resistance to abiotic stress [5]. Abscisic acid is also called a stress hormone. The main function of abscisic acid is to break seed dormancy, promote seed germination, and regulate stomatal closure, root development, and resistance to abiotic stress [6].

The genomic DNA of eukaryotes is a fragmented gene that contains a large number of introns that do not encode proteins. The precursor mRNA of the initial product of transcription contains this part of the long intron, and the intron itself does not have the ability to encode a protein, therefore, this part of the intron needs to be removed. Precursor mRNA becomes mature mRNA through splicing [7]. 

Alternative splicing (AS) is the process of generating different mature mRNAs from the same mRNA precursor by selecting different combinations of splicing sites. AS produces various transcripts and diverse proteins in eukaryotic cells [8]. AS involves many biological processes in plants, especially adverse abiotic environmental stress. In adverse environments, such as drought and cold, certain genes in plants will undergo alternative splicing to respond to the living environment [9]. AS is a significant mediator of gene expression and increases proteome diversity. 

The process of AS is mainly catalyzed by a spliceosome. A spliceosome mainly contains ribonucleoproteins. The composition of U1, U2, U4, U5, and U6 is the main type of spliceosome complex at present. The spliceosome assembles at one intron and recognizes splice sites to complete alternative splicing [10]. There are eight manners of AS, and the common four are: intron retention (retained introns, RI), variable acceptor sites (alternative 3′ splicing sites, A3), variable donor sites (alternative 5′ splicing sites, A5), exon skipping (skipped exons, SE). The remaining four include alternative splicing of the transcription start site (alternative first exon, AF), alternative splicing of the transcription termination region (alternative last exon, AL) alternative exon (AE), and mutually exclusive exon (MX) [11,12]. The regulation of AS is accomplished by the recognition of specific elements in the precursor mRNA by alternative splicing regulatory proteins, namely trans-acting factors. Splicing factors include serine/arginine-rich (SR) proteins and heterogeneous nuclear ribonucleoprotein (hnRNP) family proteins, RNA helicases, kinases, and many other factors [13]. Cis-regulatory RNA sequences can be further divided into enhancers that promote splicing at specific splice sites and splicing silencers that reduce splicing. Alternative splicing is a mechanism for regulating gene expression at the mRNA level [14].

Recently, sequencing technology has been widely used in plant genome sequencing and transcriptome sequencing. It has been found that alternative splicing has a profound impact on plant growth and abiotic stress [11]. AS is very widespread in eukaryotes. It has been demonstrated that in mammals, over 95% of genes undergo intron-containing transcription. Research now shows that in plants such as *Arabidopsis*, rice, and maize, 60% of genes that contain introns are alternatively spliced [13,15,16]. More transcriptome data and RNA-seq data have shown that the AS of genes can be induced by environmental conditions [17,18,19]. In the research, the AS pattern of 9200 genes changed in response to heat stress in tomatoes, while 1000 genes had differential splicing under cold-stressed conditions [20,21,22]. The gene expression induced by abiotic stress was more prone to AS. Moreover, abiotic stresses imposed by high temperatures induced different splice sites [23,24].

The promoters of ABA-responsive genes contain cis-acting elements ABA-responsive element (ABRE) and a coupling element (CE). ABRE-binding factors (AREB/ABF) involved in ABA-dependent stress responses. As the AREB/ABF subfamily, AREB2 exists in five different isoforms by the use of splice acceptor sites in exons 4. Isoform1 and isoform3 represent the major and minor forms, respectively. Isoform3 is expressed in root differentiation and elongation [9,25,26]. Sixteen transcripts are produced by 13 PYL ABA receptor genes in maize B73, of which two genes undergo alternative splicing. Nineteen transcripts generate from 14 A-type *PP2C* genes in maize B73, four of which undergo alternative splicing. There are thirty-two transcripts from the 12 *SnRK2* genes in maize B73, 10 of which undergo alternative splicing [27]. Additionally, HYPERSENSITIVE TO ABA1 (*HAB1*) encodes PP2Cs. The PP2Cs negatively regulate abiotic stress via mediating ABA signaling. The splicing of *HAB1* can be controlled by the splicing factor, RBM25, by RNA-seq [28]. *HAB1* produces four transcripts via AS [28]. The first two transcripts are involved in seed germination by mediating ABA signaling [28]. The first transcript, *HAB1.1*, inhibited is the OST1 protein kinase by interacting with the OST1 protein kinase. The second transcript, *HAB1.2*, which retains the third intron, formed is a truncated protein. *HAB1.2* positively regulates ABA signaling by interacting with OST1 [9]. Pladienolide B (PB) is involved in the splicing and ABA response. PB affects the splicing of PP2C mRNAs, which reduce the activity of PP2Cs [29]. However, PB mediates SnRK2.6 activity by binding amino acid residue to Leu-46 [30].

Alternative splicing is widely involved in rapidly regulating plant developmental and environmental changes [31]. When the external environment changes, such as high temperatures, drought, and other harsh conditions, plants can improve their ability to resist stress through alternative splicing of their own transcriptomes. Furthermore, ABA is an important phytohormone in response to various stress signals. Research shows that both the ABA signaling pathway and AS mediate plant growth and abiotic stress [30,32,33]. However, how ABA regulates plant development and stress tolerance through AS is still unclear. In this review, we discuss that ABA mediates plant abiotic stress tolerance and development, particularly by alternative splicing events.

## 2. ABA Mediates Plant Development and Abiotic Stress

### 2.1. ABA in Responses to Abiotic Stress

Abiotic stress seriously endangers crop production around the world. Abiotic stresses usually lead to the rapid accumulation of ABA in plants, which can cause a large number of cellular and molecular responses [34]. The main functions of ABA are: (1) to make seeds dormant and delay seed germination. (2) to induce stomatal closure. Its most important function is to regulate the water potential balance and osmotic pressure balance of cells. Environmental stresses activate the expression of the ABA biosynthesis gene. The 9-cis-epoxy compound that catalyzes this reaction, carotene dioxygenase (*NCED*), encodes the ABA synthesis enzyme in higher plants. Hai et al. observed that the overexpression of *NCED3* in plants significantly increased the ABA content and improved salt tolerance [35]. Research shows that *NCED3* responds to water deficits by regulating ABA synthesis [36]. ABA levels in plants are regulated by both synthesis and decomposition. *CYP707A* plays a dominant role in abscisic acid catabolism, and it was demonstrated in *Arabidopsis* that a rapid increase in *CYP707A* mRNA levels is closely related to a sharp decrease in abscisic acid levels [37]. Therefore, both promoting the synthesis of ABA and inhibiting the decomposition of ABA can increase the level of ABA in plants. 

The regulatory network of abscisic acid signaling is very complex, and its key regulatory mechanism lies in the interaction of its core components. The core members of the ABA signal are usually three components, which are the PYR/PYL/RCAR receptors (PYLs), protein phosphatase 2c (PP2C), and sucrose nonfermenting 1-related protein kinase 2s (SnRK2s) [38]. When plants encounter abiotic stress, ABA inhibits PP2Cs by binding to the PYR/PYL, and PP2C phosphatases bind to dephosphorylates in SnRK2s protein kinases, and then the SnRK2s phosphorylate several transcription factors and membrane proteins involved in the ion channels [39]. The AREB/ABF transcription factors (TFs) mediate the expression of stress-responsive genes to increase plant tolerance [40,41]. AREB/ABFs and ABSCISIC ACID-INSENSITIVE5 (ABI5) are the main transcription factors that regulate ABA-mediated plant growth and adaptation to the external environment.

SnRK2s phosphorylate AREB/ABFs to activate the expression of ABA-responsive genes. In *Arabidopsis*, the AREB/ABFs family contains three members: AREB1/ABF2, AREB2/ABF4, and ABF3/DPBD5 [42]. The phosphorylated AREB regulates the expression of ABA-responsive genes by binding to ABRE cis-elements. The AREB/ABFs can regulate the expression of *atRD29B*, *atAIL1*, *atEM1*, *atEM6*, *atRAB18*, and other LEA-like genes in response to cold and dehydration stress [43]. The AREB/ABFs can regulate the expression of *atAHG1*, *atAHG3*, *atHAI1*, *atHAI2*, *atHAI3*, and other PP2C genes in response to drought stress. The AREB/ABFs can regulate expression of *atADF5*, *atTPPE*, and *atTPPI* genes to mediate stomatal in response to abiotic stress, and regulate the expression of *atHsfA6A*, *atHsfA6B*, and *atMYB102* genes in response to osmotic stress [44]. In addition, PP2Cs genes mainly contain *ABI1*, *ABI2*, *HAB1* and *HAB2.* The SnRK2 kinases mainly include SnRK2.2, SnRK2.3, and SnRK2.6 [45,46,47]. SnRK2s phosphorylate the membrane proteins of ion channels. The ion channels, such as the potassium channel KAT1 and the slow anion channel SLAC1, regulate guard cells [48,49,50]. Guard cells improve plant resistance to abiotic stress mainly by regulating guard cell ion channels [51].

ABA functional mutants show marked insensitivity to all aspects of ABA response, especially resistance to stressful environments. The expression of the ABA receptor *msPYL* gene is increased under low-temperature stress [52]. *PP2C* genes are involved in salt stress and fruit development [46,53]. The *CBF* gene is highly expressed in grape tissues under low-temperature stress and ABA. ABA and low temperature have synergistic effects on the activation of these transcription factors and simultaneously affect the increase of shoot cold tolerance and antioxidants and dehydrin gene expression. Alkaline stress can cause the accumulation of reactive oxygen species, which is the main cause of root damage and seedling wilting. Salt stress leads to the accumulation of ABA in plant roots, which is transmitted upward through the xylem, regulates stomatal closure, and reduces plant transpiration and water loss.

Stomatal closure is considered one of the ABA responses to abiotic stress [54]. SMALL AUXIN UP-REGULATED RNA32 (SAUR32) encodes the *AtSAUR32* gene, which is induced by ABA. The AtSAUR32-overexpressed line increases the tolerance of drought stress by stomatal closure [55]. Studies have shown that the transcriptome sequencing of ABA- and salt-stressed tobacco had similar differential gene expressions in many ways [56]. The *msPYL* gene of the ABA receptor may be a response to cold stress by the bioinformatics methods in alfalfa (*Medicago sativa* L.) [52]. 

The accumulation of ABA in plants has a circadian rhythm and peaks in the afternoon to evening. The results of chromatin immunoprecipitation sequencing (ChIP-seq) showed that the early morning component of the circadian clock LHY could bind to the promoter regions of genes related to ABA synthesis and signaling pathways, regulating their transcription [57]. EEL (ENHANCED EM LEVEL) is a transcription factor that responds to ABA during plant dehydration, and GI binds to the promoter region of *NCED3* by forming a complex with EEL, promoting the expression of NCED3 [58]. The expression of *NCED3* enhances drought tolerance via the regulation of abscisic acid synthesis. The accumulation of ABA is lower in the *gi-1* mutant than that of Col-0. The degree of stomatal closure in the *gi-1* mutant is weakened during the dehydration process, resulting in an increase in the rate of leaf water loss and a decrease in plant survival [58].

In addition, Epigenetic regulation helps plants resist abiotic stresses. The ABA-dependent gene may confer drought stress tolerance due to CG and CHG hypomethylation changes in maize [27]. DNA methylation of many stress-responsive genes regulates ABA-mediated abiotic stress signaling.

### 2.2. ABA Mediates Plant Development

The role of ABA in the growth and development of plants is also multi-faceted. ABA inhibits seed germination by promoting seed drying and dormancy [59] and is involved in floral transition, lateral root formation, and seedling growth [60].

Studies have shown that at the seed maturity stage, the ABA content was higher in up-regulated ABA anabolic genes and at the germination stage, the ABA content was low-level in the up-relegated ABA catabolism gene expression [61,62]. Studies have also shown that the *ABI5* overexpression is hypersensitive to ABA treatment while the *abi5* mutant is insensitive to ABA at the germination stage [63]. The regulation of seeds by ABA is mainly achieved through the *ABI5* transcription factor, which prevents germination and post-germination growth under adverse conditions. In the ABA signaling pathway, SnRK2 kinase phosphorylates *ABI5*, and the phosphorylated ABI5 regulates the expression of downstream genes [44]. *ABI5* mediates seed germination by regulating the downstream genes, *atPGIP1* and *atPGIP2*. ABI5 mediates the stability of seed germination by regulating the downstream genes, *atPHO1* and *atCAT1* [41]. *ABI5* mediates plant resistance to abiotic stress by regulating LEA protein genes, such as *atEM1* and *atEM6. ABI5* mediates the vegetative growth of plants by regulating *atSGR1*, *atNYC1*, and *atSAG29* [64].

Flowering is another important biological process for plants. Plant flowering is usually regulated by internal signals and the external environment. The flowering time is affected by the synthesis and signal transduction of ABA, in which ABA usually inhibits flowering. The regulation of the flowering transition by ABA is mainly achieved by the ABI5 transcription factor. In the ABA signaling pathway, SnRK2.6 kinase phosphorylates ABI5, and the phosphorylated ABI5 binds to the ABRE cis-acting element in the *atFLC* promoter region to regulate the expression of *atFLC* and delay the flowering transition [65]. In the ABA signaling pathway, ABI5 binds to the *atSOC1* promoter region and induces the expression of *atSOC1* to promote flowering [41]. Studies have shown that ABSCISIC ACID-INSENSITIVE4 (*ABI4*)-overexpression delayed the flowering time while the early flowering phenotype of the *abi4* mutant [66]. This suggests that *ABI4* negatively regulates plant flowering time. A schematic diagram of the transcriptional regulation of ABA-mediated plant abiotic stress and development is summarized in Figure 1.

## 3. ABA Mediates Plant Development and Abiotic Stress via Alternative Splicing

Numerous transcriptome results show that ABA can significantly affect numerous and key transcription factors through the transcriptional regulatory pathway of signal transduction by regulating the expression of a large number of genes [58]. Studies have shown that ABA-induced differentially expressed genes have little intersection with differentially alternatively spliced genes, suggesting that the mechanism by which ABA regulates alternative splicing to help plants adapt to the environment is an independent mechanism [29,67]. At present, the mechanism of ABA regulation of AS has not been clearly elucidated.

Widespread use of RNA-seq data provides a favorable resource for analyzing AS. In the ABA treatment group and the ABA control group at the seed germination stage of *Populus hopeiensis*, the proportion of the RI differential variable splicing events was still the highest, accounting for 43.16%, the A3 event type accounted for 24.01%, A5 event type accounted for 13.98%, SE event type accounted for 8.51%, AF event type accounted for 7.90%, AL event type accounted for 2.43%, and MX type events did not have differential variable splicing events [68]. By analyzing the transcriptome sequencing results of ABA-treated *Nicotiana tabacum*, the SE event type was the highest proportion, followed by RI and A3, also occupying a relatively high proportion; MEX was the lowest proportion [58]. Through the results of the differential alternative splicing analysis of ABA-treated *Arabidopsis* sequencing, we found that the proportion of the AL differential alternative splicing events in the ABA-treated group was the highest compared with the control group, accounting for about 35.4%. Followed by the AF event types, the proportion of variable splicing was 34.9%, the proportion of AE events was 14%, the proportion of RI events was 10%, and the proportion of other variable splicing was 10%. The type events, SE and MX, accounted for very little, both less than 10% [67].

From the above, ABA treatment was shown to induce dramatic changes in AS in many dicot species. ABA treatment not only increased the total number of genes with AS but also increased the percentage of genes with AS at the transcriptional level. Different types of ABA-induced differential alternative splicing events occur in different plants, but the most predominant type of splicing event is RI. We speculate that RI-type alternative splicing events play a certain biological role in Hu plants affecting environmental complexity and adaptability, while MX-type events may not play a major regulatory role in this regard. Differential splicing types (DAS) induced by abscisic acid in different plants are shown in Table 1.

ABA-induced alternative splicing is mostly derived from transcriptome sequencing results. The functional significance of most AS events regulated by ABA in plants has not been demonstrated. The ABA-induced alternative splicing of genes in the ABA signaling pathway in *Arabidopsis* is summarized in Table 2.

ABA-induced AS occurs primarily in regulatory genes, such as transcription factors, protein kinases, and splicing factors. Splicing factors have long been shown to be involved in spliceosome assembly and AS site selection; in particular, the U1snRNA and U2AF are major players in 5′ and 3′ splice site recognition, respectively [69,70]. ABA-induced differential alternative splicing is involved in ABA responses by increasing the number of unconventional splicing sites. Changes in increased splice sites under ABA treatment may be due to the recruitment of different splicing factor isoforms in the splicing complex. 

SR genes are an important part of the splicing complex, which are involved in splicing mainly by co-recognizing the 5′ cleavage site with U1snRNA and U1-70K. The splicing mode of SR genes can be affected by various abiotic stresses and hormones. Among them, ABA can induce the splicing of most SR genes, which is summarized in Table 3. Among the several affected SR genes, the genes that can receive ABA that significantly affect the splicing pattern are *SR34*, *SR34b*, *RS31a*, and *SCL33* [67,71]. However, the functional significance of the alternative splicing products of SR genes is currently unclear. The current study shows that both ABA and abiotic stress can express the SR gene, thereby affecting the splicing of its own and other genes [72].

Among them, SR45, as a spliceosome protein, interacts with the U1-70K protein to play a splicing function. ABA regulates the splicing of the key genes of ABA signaling, such as *HAB1*, by mediating the expression of SR45, thereby regulating salt stress and drought stress; at the same time, ABA can further mediate the splicing of SRP30 and SR34 by regulating the expression of SR45, thereby affecting a series of gene splicing, regulating flowering and root development in *Arabidopsis* [73,74]. Studies have shown that ABA may mediate gene splicing by regulating the coupling of SR proteins to protein kinases and transcription factors, thereby regulating plants [75]. The nuclear distribution of *Arabidopsis* SR proteins is affected by abiotic stress. Therefore, we speculated that the effect of ABA on splicing might act through the distribution of SR proteins [76]. However, the mechanism by which ABA mediates alternative splicing through SR genes remains to be elucidated. 

Alternative splicing factors typically responded to ABA treatment. *U2AF35A* and *U2AF65A/B* are important subunits that help recruit U2 small nuclear ribonucleoproteins (snRNP). ABA not only regulated and induced the transcription of U2AF but also affected its splicing. Alternatively, *U2AF* regulated the splicing of key flowering genes. The splicing factor *AtU2AF65B* mediated ABA-mediated flowering by regulating the AS of *ABI5* and *FLC* [70]. *ABI5* regulated ABA-mediated seed germination and flowering transition. *ABI5* regulated floral transition by activating the expression of *FLC* [70]. The splicing efficiencies of *FLC* and *ABI5* were regulated by ABA in the *atU2AF65B* mutants, which indicates that *atU2AF65B* could regulate ABA-mediated flowering [70,77]. *AtU2AF65B* regulated flowering by mediating the splicing of *MAF1* (AT1G77080, flowering regulator) and *MAF2* (AT5G65050, flowering regulator) [78]. Additionally, U2AF65B interacted with U2AF35 to mediate the splicing of *HAB1* (AT1G72770, ABA signal regulator) and DWA3 (AT1G61210, ABA signal regulator), affecting salt and drought stress [79].

LUC7 is a subunit of U1 snRNP in plants that respond to both ABA and abiotic stresses. LUC7 interacts with U1-70K to participate in the splicing function. Studies have shown that ABA can regulate cold and salt stress by affecting the expression of the splicing factor, LUC7, and then regulating the alternative splicing of the stress genes *ACA4* (AT2G41560, salt stress regulator) and *K9L2.5* (AT5G44290, cold stress regulator). ABA can mediate the development of plant stems and leaves by regulating the splicing of genes *LUH* (shoot and leaf development regulator) and *TRM3* (shoot and leaf development regulator) by regulating the expression of LUC7 [80]. 

The Sm-like protein (LSm) is the core component of U6 RNPs and is involved in pre-mRNA splicing [9]. LSm5 is encoded and is super sensitive to ABA and the drought 1 gene (SAD1). *sad1* mutants are sensitive to the ABA inhibition of germination and root growth, and SAD1 depletion showed canonical 5’ and 3’SS recognition. SAD1-OE plants exhibited more salt tolerance via strengthening the AS of salt-induced genes and the recognition accuracy [81]. LSM5 interacts with LSM6/7 to participate in splicing. LSM5 regulates the salt stress resistance of *Arabidopsis* by regulating the splicing of stress-responsive genes, *SnRK2.1/2.2* (AT5G08590, osmotic stress regulator), *SOS2* (AT5G35410, salt stress regulator), and *DREB2A* (AT5G05410, drought stress regulator). In addition, LSM5 regulates *Arabidopsis* root development by regulating the splicing of *AUX1* (AT2G38120, root development regulator) and *EIN2* (AT5G03280, root development regulator) genes.

STABILIZED1 (*STA1*) encodes a nuclear protein and interacts with U5-snRNP. STA1 responds to ABA and abiotic stress by modulating splicing. STA1 is important for the heat shock transcription factor (*HSFA3*) and *COR15A* expression [82,83]. U5 snRNP normally interacts with U4/6 snRNP to mediate splicing. U5 snRNP responds to external temperature stress by regulating the splicing of *COR15A* (AT2G42540, cold stress regulator) and *HSFA3* (AT5G03720, heat stress regulator). However, it is currently unclear how U5 snRNP regulates splicing to mediate root development.

FERONIA (FER), a receptor-like kinase, is a receptor for the rapid alkalinization factor 1 (RALF1) peptide. FER and RALF1 form a complex. The complex interacts with and phosphorylates RNA-binding proteins, which triggers AS [84]. Among them, the glycine-rich RNA binding protein 7 (GRP7) is a splicing factor and regulates seed germination, seedling growth, and flowering [85]. In our study, GRP7 phosphorylation enhanced its interaction with U1-70K, modulating the AS in response to external stimuli and ABA signaling [84]. Both GRP7 and FER are involved in ABA signaling. GRP7 regulated drought resistance in plants by mediating the splicing of *ABF1* (AT1G49720, ABA signal regulator). In addition, GRP7 regulated the root development of plants by mediating the splicing of *CUB9* (AT3G07360, root development regulator).

RBM25 is a splicing factor that responds to salt, drought, and osmotic stress. RBM25 affects plant aridity by regulating ABA-mediated stomatal movement. RBM25 normally interacts with LSM6/7 for splicing. RBM25 responds to osmotic stress by affecting *HAB1* splicing. RBM25 responds to plant stem and leaf development by affecting the splicing of *PAC* (AT2G48120, leaf development regulator) and *PSBQA-2* (AT4G05180, photosynthesis regulator). In summary, a simplified mechanism diagram of ABA mediating abiotic stress and plant development by affecting splicing is shown in Figure 2.

ABA-induced alternative splicing is mostly derived from results of the transcriptome sequencing. The functional significance of most AS events regulated by ABA in plants has not been demonstrated. The following studies have shown that ABA-induced splicing of key genes plays an important role in plant adaptation to the environment. Sanyal et al. showed that ABA treatment significantly enhanced the induction of five splice variants of *CIPK3*. CIPK3 is a protein kinase, which is a serine-threonine protein whose expression increases in response to abscisic acid, cold, drought, high salt, and trauma conditions. *CIPK3* regulates stress and development by participating in Ca^2+^ signaling and ABA pathways. Among them, *CIPK3.1* and *CIPK3.4* splice variants were the two most dominant transcripts under the ABA treatment, while *CIPK3.1* showed an interaction with ABR1 in response to ABA, osmotic stress, and drought stress [79,86]. The study by Zhan et al. showed that ABA has an effect on the splicing pattern of *Arabidopsis* wild-type, which can significantly induce the alternative splicing of 27 genes. Among them, the *AtSPL2* gene is involved in shoot maturation at the reproductive stage. *AtATERDJ3B* is involved in protein folding, and *AtNUP50* is involved in intracellular transport [29]. ABA mediates plant development and abiotic stress responses by affecting splicing of genes, as shown in Table 4.

AS plays an important regulatory role in plant cells, and AS can regulate the transcriptional level by introducing premature termination codons (PTC). ABA-induced alternative splicing is often accompanied by a shift of the reading frame, resulting in premature codon termination. Most aberrantly spliced RNAs are degraded through the NMD (nonsense-mediated decay/nonsense-mediated degradation) pathway. The NMD pathway can prevent abnormally spliced RNAs from being translated into abnormal proteins, thereby inhibiting abnormal biological effects in plants. AS produces truncated non-functional proteins through RI or PTC to adjust the amounts of functional proteins in plants in time so that plants have stronger adaptability. 

AS is an important mechanism for altering the transcriptome in response to the environment and improving reproduction. Dysregulation of alternative splicing is a hallmark feature of plant adaptation. Studies have shown that there is little intersection between ABA-induced differentially expressed genes and key differentially spliced genes [67]. The stress-induced transcriptome results of wheat and rice are consistent with the results [16,87]. This suggests that AS is an independent mechanism by which ABA regulates plant growth and environmental responses. The ABA-induced alternative splicing mechanism will be a potential target for improving the ability of plants to adapt to the environment. The study of the ABA-induced variable shearing mechanism will offer an important mechanism for crops to adapt to the environment and improve crop quality through the variable shearing mechanism in the future, and, at the same time, make a strong guarantee for future food security.

## 4. Conclusions

ABA is an important hormone in response to plant growth and abiotic stress. There are different mechanisms by which ABA affects the environment. First, from the transcriptional level, ABA regulates the up-regulation or down-regulation of a large number of genes by regulating kinases and a large number of transcription factors in the signaling pathway, thereby increasing the abundance of proteins and improving the adaptability of plants. Secondly, from the post-transcriptional level, ABA affects the splicing of its own and other genes by regulating the expression of splicing factors, such as SR genes and U2AF, to produce different splicing isoforms, thereby increasing the abundance of proteins and improving plant adaptation. 

## Figures and Tables

**Figure 1 ijms-23-03796-f001:**
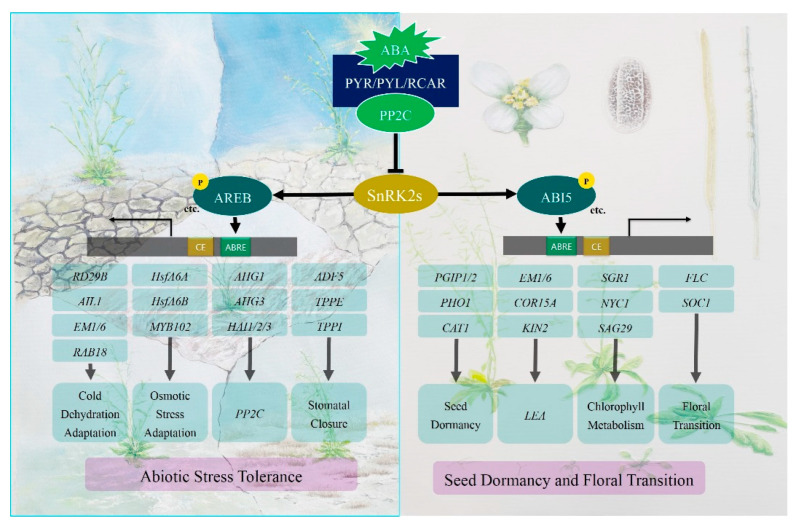
Schematic diagram of ABA-mediated transcriptional regulation of plant. The PYL-PP2C-SnRK2s-TFs complex regulate the ABA-mediated ABRE-dependent gene expression. Two distinct family TFs (ABI5 and AREB/ABF) regulate the ABA-mediated gene expression in response to abiotic stress and development in *Arabidopsis* by interacting with cis-elements in the upstream promoter regions of target genes. See text for details.

**Figure 2 ijms-23-03796-f002:**
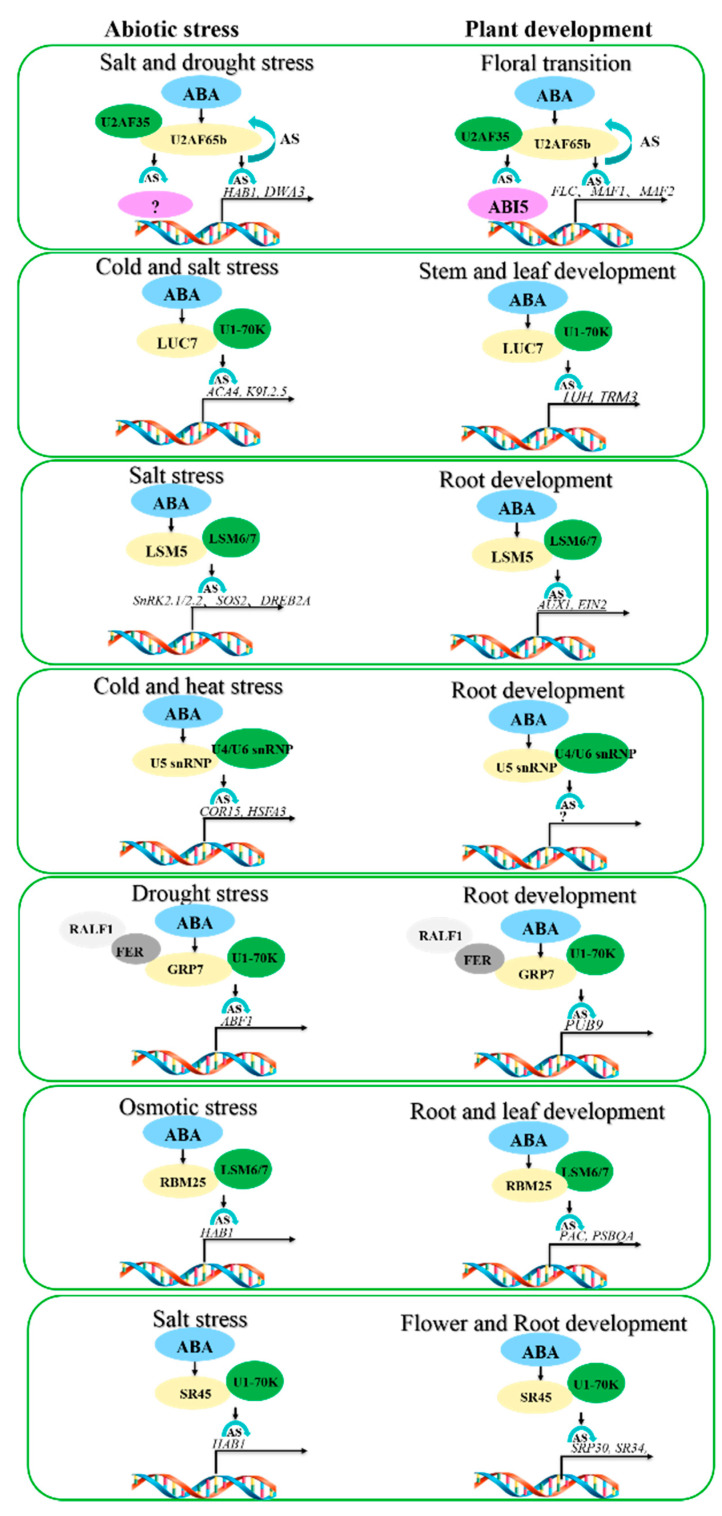
ABA mediates abiotic stress and plant development by affecting the expression or splicing of the same splicing factor, thereby affecting the splicing of different genes. Orange ovals represent splicing factors acted on by ABA, green ovals represent interacting splicing factors, purple ovals represent transcription factors, white ovals represent alkalizing factors, and gray ovals represent kinases.

**Table 1 ijms-23-03796-t001:** Differential splicing type (DAS) statistics of different plants induced by abscisic acid.

Plants	Major DAS Type	References
*Arabidopsis*	AL > AF > AE > RI > SE > MX	[67]
*Nicotiana tabacum*	SE > RI > A3 > A5 > MX	[56]
*Populus hopeiensis*	RI > A3 > A5 > SE > MX	[68]

**Table 2 ijms-23-03796-t002:** Summary of the alternative splicing of genes in the ABA-induced abscisic acid (ABA) signaling pathway in *Arabidopsis thaliana* [67].

Locus	Gene	Transcript Type
AT1G17550	*HAB2*	*HAB2-iso1,2,3*
AT1G45249	*ABRE1*	*ABRE1-iso1,2,3*
AT4G34000	*ABF3*	*ABF1-iso1,2,3*
AT5G25610	*RD22*	*RD22-iso1,2*
AT4G27410	*RD26*	*RD26-iso1,2,3,4*
AT1G20620	*CAT3*	*CAT3-iso1,2,3,4*
AT5G62470	*MYB96*	*MYB96-iso1,2*
AT1G20630	*CAT1*	*CAT1-iso1,2*
AT4G46270	*GBF3*	*GBF3-iso1,2,3,4,5,6*
AT4G19230	*CYP707A1*	*CYP707A1-iso1,2*
AT4G26080	*ABI1*	*ABI1-iso1,2*
AT5G57050	*ABI2*	*ABI2-iso1,2,3*

**Table 3 ijms-23-03796-t003:** Summary of ABA-induced alternative splicing events of *Arabidopsis* splicing factors [67,71].

Locus	Splicing Factors
AT1G02840	*SR34*
AT4G02430	*SR34b*
AT1G09140	*SR30*
AT3G61860	*RS31*
AT2G46610	*RS31a*
AT4G25500	*RS40*
AT5G52040	*RS41*
AT2G24590	*RSZ22A*
AT3G53500	*RSZ32*
AT2G37340	*RSZ33*
AT5G64200	*SC35*
AT1G55310	*SCL33*
AT3G55460	*SCL30*
AT3G13570	*SCL30A*
AT1G16610	*SR45*
AT3G50670	*U1-70K*
AT1G27650	*U2AF35A*
AT1G60900	*U2AF65B*
AT2G43810	*LSM6B*

**Table 4 ijms-23-03796-t004:** Summary of the fundamental studies on AS in ABA-mediated plant development and abiotic stress responses.

Gene ID	Gene Name	AS Mediated Development and Abiotic Stress	References
AT2G26980	*AtCIPK3*	ABA, osmotic stress, and drought stress	[79,86]
At5g43270	*AtSPL2*	shoot maturation	[29]
At3g62600	*AtATERDJ3B*	protein folding	[29]
At3g15970	*AtNUP50*	intracellular transport	[29]

## Data Availability

Not applicable.

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
