# Peer review of "ABA Mediates Plant Development and Abiotic Stress via Alternative Splicing"

_ijms, 2022, doi:10.3390/ijms23073796_

Round 1
Reviewer 1 Report
The review manuscript entitled with “ABA mediates plant development and abiotic stress via alternative splicing” was studied by Yang et al. The authors provided the recent studies and attempts in understanding the crosstalk between ABA and alternative splicing under stress and non-stress condition. The topic seems to be interesting. However, the manuscript needs to be improved and I recommend several comments below.
- The authors can improve the abstract and conclusion part by providing more general information and perspectives of ABA and alternative splicing.
- The letter of “Arabidopsis” should be replaced by italic letters in the manuscript (i.e line 57, 96…)
- In line 76, “on the other hand” should be started with a capital letter.
- In line 91, “Han et al” should be corrected to “Hai et al”.
- In line 141, “Col0” should be corrected to “Col-0”.
- In line 155, please put the punctuation at the end of sentence.
- In line 232 and, “Rice gene OsHsf24 were also…” should be corrected to “Rice gene osHsf24 was…”.
- In line 259, “Sad1” should be replaced with an italic letter.
- In the reference, please check the reference format of the Journal. The style is not uniformed as journal recommended.
- Overall, the English in manuscript should be grammatically and technically improved by a professional.
Author Response
The review manuscript entitled with “ABA mediates plant development and abiotic stress via alternative splicing” was studied by Yang et al. The authors provided the recent studies and attempts in understanding the crosstalk between ABA and alternative splicing under stress and non-stress condition. The topic seems to be interesting. However, the manuscript needs to be improved and I recommend several comments below.
Response: Thank you for your careful check and valuable comments. We feel so sorry for the inconvenience brought to you. The mistakes have been corrected in the revised manuscript. Revised sections are marked in blue and the revised manuscript has been submitted. We are very sorry for our carelessness.
Comments: 1. The authors can improve the abstract and conclusion part by providing more general information and perspectives of ABA and alternative splicing.
Response: We gratefully appreciate for your valuable comment. We have revised the abstract and conclusion part according to your suggestions. Thank you so much for your careful check. We feel so sorry for our carelessness.
Comments: 2. The letter of “Arabidopsis” should be replaced by italic letters in the manuscript (i.e line 57, 96…)
Response: Sorry for our mistakes. We have italicized the letters "Arabidopsis" in the manuscript. And we have checked these similar errors thoroughly in whole text and revised them.
Comments: 3. In line 76, “on the other hand” should be started with a capital letter.
Response: We feel so sorry for our carelessness. We have corrected this the detail information according to your suggestions (shown in lines 84). Thank you so much for your careful check.
Comments: 4. In line 91, “Han et al” should be corrected to “Hai et al”.
Response: Thank you so much for your careful check. “Han et al” have been corrected to “Hai et al” (shown in lines 111). We feel so sorry for our carelessness.
Comments: 5. In line 141, “Col0” should be corrected to “Col-0”.
Response: Thank you for your comment. “Col0” have been corrected to “Col-0” (shown in lines 157). Thank you so much for your careful check.
Comments: 6. In line 155, please put the punctuation at the end of sentence.
Response: We feel so sorry for our carelessness. We have done revisions to put punctuation at the end of the sentence (shown in lines169). Thank you so much for your careful check.
Comments: 7. In line 232 and, “Rice gene OsHsf24 were also…” should be corrected to “Rice gene osHsf24 was…”.
Response: Thank you so much for your careful check. “Rice gene OsHsf24 were also…” have been corrected to “Rice gene osHsf24 was…” (shown in lines 267). We feel so sorry for our carelessness.
Comments: 8. In line 259, “Sad1” should be replaced with an italic letter.
Response: Sorry for our mistakes. “Sad1” have been replaced with an italic letter (shown in lines 253). Thank you so much for your careful check.
Comments: 9. In the reference, please check the reference format of the Journal. The style is not uniformed as journal recommended.
Response: Thank you so much for your careful check. The format of the references has been uniformed according to the requirements of the journal in the revised manuscript. We feel so sorry for our carelessness.
Comments: 10. Overall, the English in manuscript should be grammatically and technically improved by a professional.
Response: Thank you so much for your careful check. The strong English revision have been corrected in the revised manuscript. We have tried our best to make our expressions more accurate and standardized, and easier for everyone to understand. We feel so sorry for our carelessness.
Reviewer 2 Report
The theme addressed by the manuscript entitled "ABA mediates plant development and abiotic stress via alternative splicing" is extremely relevant and current. However, the manuscript needs a significant improvement in the flow of ideas, as well as the current state of knowledge about the theme, the gaps and the future perspectives to be addressed. In many places authors just wrote some general sentences not mining deeply the exact mechanism how ABA mediated the plant development and stress tolerance by alternative splicing. What is the role of ABA in alternative splicing? How it will benefit the agriculture sector and future food safety? Beside this, the paper lacks a good graphical presentation and conclusion. The conclusion is too simple and not up to the point. Therefore, the paper cannot be accepted in its present form.
Author Response
The theme addressed by the manuscript entitled "ABA mediates plant development and abiotic stress via alternative splicing" is extremely relevant and current. However, the manuscript needs a significant improvement in the flow of ideas, as well as the current state of knowledge about the theme, the gaps and the future perspectives to be addressed. In many places authors just wrote some general sentences not mining deeply the exact mechanism how ABA mediated the plant development and stress tolerance by alternative splicing. What is the role of ABA in alternative splicing? How it will benefit the agriculture sector and future food safety? Beside this, the paper lacks a good graphical presentation and conclusion. The conclusion is too simple and not up to the point. Therefore, the paper cannot be accepted in its present form.
Response: We feel so sorry for the inconvenience caused to you. We greatly appreciate your careful review of the manuscript and insightful valuable comments. Based on your suggestions, we have revised the manuscript to the greatest extent possible. The revised part of the manuscript is marked in blue font, and the revised version has been submitted to the system.
Below, I will respond based on your question.
- What is the role of ABA in alternative splicing?
Response:
First, ABA affects the splicing of other genes, including its own splicing, by affecting the expression of splicing protein factors. Current studies have shown that the expression of one SR protein alters the splicing of its own pre-mRNA and other SR genes, and ABA has a regulatory effect on SR protein expression.
Second, phosphorylation of splicing proteins is critical for spliceosome assembly and cleavage site selection. While stress can affect the phosphorylation state of splicing proteins, it is speculated that ABA may affect the splicing of itself and other proteins by affecting the phosphorylation of splicing proteins.
The distribution of splicing factors is affected by stress and phosphorylation. Therefore, it is speculated that the effect of ABA on splicing may be the result of changes in the distribution of splicing factors in the nucleus.
- How it will benefit the agriculture sector and future food safety?
Response:
First of all, the challenges of the global extreme environment for crop growth and the current situation of the world population is still growing and the demand for food production, it is more and more important to improve plant growth and resistance at the molecular level to improve crop yield and quality.
ABA is an important plant hormone that regulates plant growth and development and stress resistance. It has been well understood that ABA regulates transcription factors at the transcriptional level to regulate gene expression.
Second, transcriptome data analysis revealed that there was little crossover between ABA-induced Arabidopsis differentially expressed genes and key differentially spliced genes. The same is true for the stress-induced transcriptomes in wheat and rice. This suggests that AS is an independent mechanism by which ABA regulates plant growth and environmental responses.
Then, the ABA-induced alternative splicing mechanism would be a potential target for improving the ability of plants to adapt to the environment. The study of ABA-induced variable shearing mechanism will lay an important mechanism for future crops to adapt to the environment and improve crop quality through the variable shearing mechanism, and at the same time provide a strong guarantee for future food security.
- Beside this, the paper lacks a good graphical presentation.
Response: We feel so sorry for the inconvenience caused to you. We have done our best to modify the graph as shown below.
- The conclusion is too simple and not up to the point.
Response: According to your suggestion, the concluding section of the manuscript has been rewritten. The conclusion part is as follows:
ABA is an important hormone in response to plant growth and abiotic stress. There are different mechanisms by which ABA affects the environment. First, from the transcriptional level, ABA regulates the up-regulation or down-regulation of a large number of genes by regulating kinases and a large number of transcription factors in the signaling pathway, thereby increasing the abundance of proteins and improving the adaptability of plants. Secondly, from the post-transcriptional level, ABA affects the splicing of its own and other genes by regulating the expression of splicing factors such as SR genes and U2AF to produce different splicing isoforms, thereby increasing the abundance of proteins and improving plant adaptation.
Round 2
Reviewer 2 Report
Thank you for revising the MS based on my comments. However, the paper still lacks a good/sufficient graphical presentation. It is important to add mechanistic diagrams for section 2 and 3. This will help improve the overall quality of the paper.
Round 3
Reviewer 2 Report
Thank you for revising the MS. The MS is now in acceptable form.